# Psychological Approaches for the Integrative Care of Chronic Low Back Pain: A Systematic Review and Metanalysis

**DOI:** 10.3390/ijerph19010060

**Published:** 2021-12-22

**Authors:** Giorgia Petrucci, Giuseppe Francesco Papalia, Fabrizio Russo, Gianluca Vadalà, Michela Piredda, Maria Grazia De Marinis, Rocco Papalia, Vincenzo Denaro

**Affiliations:** 1Department of Orthopaedic and Trauma Surgery, Campus Bio-Medico University of Rome, 00128 Rome, Italy; g.petrucci@unicampus.it (G.P.); g.papalia@unicampus.it (G.F.P.); g.vadala@unicampus.it (G.V.); r.papalia@unicampus.it (R.P.); denaro@unicampus.it (V.D.); 2Research Unit Nursing Science, Campus Bio-Medico University of Rome, 00128 Rome, Italy; m.piredda@unicampus.it (M.P.); m.demarinis@unicampus.it (M.G.D.M.)

**Keywords:** low back pain, cognitive behavioral therapy, mindfulness-based stress reduction, depression, disability, fear-avoidance beliefs

## Abstract

Chronic low back pain (CLBP) is the most common cause of disability worldwide, affecting about 12% to 30% of the adult population. Psychological factors play an important role in the experience of pain, and may be predictive of pain persistence, disability, and long-term sick leave. The aim of this meta-analysis was to identify and to describe the most common psychological approaches used to treat patients who suffer from CLBP. A systematic search was performed on PubMed/MEDLINE and Cochrane Central. Overall, 16 studies with a total of 1058 patients were included in the analysis. Our results suggest that cognitive behavioral therapy (CBT) and mindfulness-based stress reduction (MBSR) interventions are both associated with an improvement in terms of pain intensity and quality of life when singularly compared to usual care. Disability also improved in both groups when compared to usual care. Significant differences in fear-avoidance beliefs were noted in the CBT group compared to usual care. Therefore, psychological factors are related to and influence CLBP. It is crucial to develop curative approaches that take these variables into account. Our findings suggest that CBT and MBSR modify pain-related outcomes and that they could be implemented in clinical practice.

## 1. Introduction

Chronic low back pain (CLBP) is the most common cause of disability worldwide [1,2], affecting about 12% to 30% of the adult population [3,4]. It is estimated that 50% to 80% of adults feel at least one episode of back pain during their lifetime [5]. Therefore, managing CLBP becomes crucial for both individuals and health care systems [1]. Chronic pain has a multidimensional nature and in addition to nociceptive and physiological aspects, it also includes aspects relating to the emotional and cognitive sphere [6]. Low back pain pathogenesis can also be diverse, including organic, non-specific etiology, and psychological causes [7,8]. Psychological factors play an important role in the experience of pain [9,10], as patients with CLBP who experience anxiety tend to exacerbate the painful sensation and increase illness behavior [11], catastrophizing pain [12,13,14]. These factors can make the pain experience, as well as the mechanical and physiological processes, last longer [15,16], causing physical and psychosocial disability [9]. In this regard, it has been shown that patients with CLBP suffering from depression experience higher levels of pain, functional disability, and lower levels of health-related quality of life (QoL) [16]. So, all psychological variables may be predictive of pain persistence, disability, long-term sick leave [11,15], significantly influencing the quality of life perceived by patients. Therefore, it is crucial to assess and address the psychological sphere as much as the other aspects, designing a holistic and integrative framework to treat patients affected by LBP [7,17]. The American Pain Society (APS) published specific evidence-based guidelines for an interdisciplinary treatment and rehabilitation (defined as an integrated intervention with rehabilitation plus a psychological and/or social/occupational component) as a treatment option for patients with chronic LBP [18]. With the advancements in health psychology, several approaches were implemented in the care of patients with chronic pain. To our knowledge, there exist different systematic reviews in literature [16,19,20,21,22] that analyze psychological approaches to treating patients who suffer from CLBP. These studies do not evaluate which approach is most used. Moreover, a comparison between different types of psychological approaches, in order to evaluate the effectiveness in terms of improvement of clinical outcomes, has not been performed. The objectives of this systematic review and meta-analysis are (1) to identify and to describe the most common psychological approaches used to treat patients who suffer from Chronic LBP, and (2) to study the effectiveness of these approaches in terms of reduction of pain, disability, fear-avoidance behaviors, anxiety, depression, and of increase in quality of life of patients with Chronic LBP.

## 2. Materials and Methods

This systematic review was performed in agreement with the Preferred Reporting Items for Systematic Reviews and Meta-Analysis (PRISMA) guidelines [23]. The protocol was previously registered on PROSPERO (registration number CRD42021255687). This review included only randomized clinical trials (RCTs) that assessed the effectiveness of the most common psychological approaches on quality of life (QoL), pain, disability and fear-avoidance behaviors in adult patients suffering from chronic low back pain (CLBP).

### 2.1. Inclusion Criteria

We included RCTs published in the last 25 years that included adult patients with CLBP; compared psychological interventions with either comparator (usual care such as health education, physical exercise, information package and waiting list); and assessed reduction of pain, disability, fear-avoidance behaviors, anxiety, depression, and increase in quality of life. Studies were excluded if they were not RCTs, if they analyzed acute or sub-acute low back pain and if they included back-surgery patients.

### 2.2. Search Methods

We performed a systematic literature search on the following databases: PubMed/MEDLINE, Scopus and Cochrane Central. No language restrictions were set. The search strategy was checked by three reviewers (G.P., G.F.P. and F.R.). We developed a specific question defining the intervention, the population, and outcomes to analyze (according with PICO method).

PICO methods. Definition of elements.

Population: the reference population included not hospitalized patients suffering from chronic low back pain. The patients included should be at least 18 years old, and they did not have to undergo surgery.Interventions: Selected psychological approachesComparison Intervention: usual care, education program, supportive care, physical exercise, physiotherapy and waiting listOutcomes: pain, disability, fear-avoidance, anxiety and depression reduction and the improvement of quality of life

The search string included the following keywords (both Mesh and free-terms in PubMed/MEDLINE): Low back pain OR “Low back pain *” OR lumbago OR “lower back pain” OR “lower back pain *” OR “Low Back Ache” OR “Low Back Ache *” OR “Low Backache” OR “Low Backache *”; cognitive behavioral therapy OR “behavioral treatment” OR “behavior treatment” OR “behavior therapy” OR “cognitive behavior treatment” OR “cognitive treatment” OR “cognitive therapy”, Mindfulness OR Meditation OR “mindfulness meditation”, “operant behavioral therapy”, hypnotism OR hypnoanalysis OR hypnotherapy, “acceptance and commitment therapy”. The reference lists of the included RCTs were examined to choose additional studies for inclusion. After removing duplicates, two reviewers (G.P. and G.F.P.) independently analyzed the abstracts. Conflicts of opinion were solved discussing with a third reviewer (F.R.). In the end, the full texts were read and checked by two reviewers (G.P. and G.F.P.), choosing the studies to include in the review and meta-analysis.

### 2.3. Data Collection, Analysis, and Outcomes

Two authors (G.P. and G.F.P.) independently extracted the following data from the studies selected: authors, year of publication, country, sample size, patients’ age and sex, intervention (s) in the experimental and in the control group, follow-up period, outcomes analyzed, tools used and conclusions.

### 2.4. Risk of Bias Assessment

Two independent reviewers (G.P. and G.F.P.) evaluated the risk of bias of the included RCTs using the Cochrane risk-of-bias tool [24]. Possible differences in the assessment were checked by a third reviewer (F.R.). Each item was classified with a low, unclear, or high risk of bias. Thus, the studies present low risk of bias in case of six or seven domains at low risk of bias, unclear risk of bias in presence of four or five domains at low risk of bias, and high risk of bias if fewer than four domains were reported at low risk of bias.

### 2.5. Statistical Analysis

A meta-analysis was produced by Review Manager (RevMan) software Version 5.4.1. Pain, disability, quality of life, depression and fear-avoidance beliefs were assessed between CBT, MBRS and control groups as continuous outcomes. In presence of different scores, the relative outcome was presented as standard mean difference (SMD) with 95% confidence intervals, while we adopted mean difference (MD) for the outcomes assessed by the same score. Instead, days without pain was calculated as a dichotomous outcome using odds ratio (OR) with 95% confidence intervals. The evaluation of the samples’ weight for this outcome was assessed by the mean value of days without pain per number of patients as events and the number of patients per number of weeks of follow-up as total. The I^2^ test was adopted to check the heterogeneity of studies included. In case of low heterogeneity (I^2^ < 55%), a fixed-effect model was used, otherwise, we adopted a random-effect model. The statistical significance of the results was set at *p* < 0.05. 

## 3. Results

### 3.1. Results of the Literature Search

The literature search yielded 3277 articles. After removal of duplicates, the reading of titles and abstracts led to 48 eligible papers. All 48 full-texts were read. Afterwards, 32 studies were eliminated for these reasons: patients who suffered from acute pain (*n* = 8), patients who suffered from sub-acute low back pain (*n* = 7), not reporting selected outcomes (*n* = 5), back surgery patients (*n* = 5), inpatients (*n* = 4), pediatric patients (*n* = 2), and hypochondriacal patients (*n* = 1). At the end of selection, 16 RCTs were included in the systematic review and meta-analysis (Figure 1).

### 3.2. Demographic Data 

The total sample consisted of 2038 adults with CLBP reviewed—1058 were in the intervention group and 980 were in the control group. Most studies were published in the USA (*n* = 8; 50%), two studies were published in UK (12.5%) and Germany (12.5%), one study was published in Italy (6.25%), in The Netherlands (6.25%), in Pakistan (6.25%) and in Sweden (6.25%). The age of the patients ranged from 40.7 to 78 years in the experimental groups, and from 40.5 to 75.6 in the control groups. The percentage of women in the studies ranged from 13% to 80% in the intervention groups and from 6% to 87% in the control groups. In Table 1 the main characteristics of included studies and samples are reported.

### 3.3. Type of Interventions

The psychological approaches most used are the cognitive behavioral therapy (CBT) and mindfulness-based stress reduction (MBSR). CBT was evaluated in eleven studies, while the remaining three studies [25,26,27] examined MBSR. Two studies evaluated both CBT and MBSR [9,28] versus usual care (Table 2). The mean follow-up was 7.8 months and ranged from 3 weeks to 15 months.

### 3.4. Clinical Outcome Data

The outcomes were analyzed by different tools. Disability was assessed using the Roland and Morris Disability Questionnaire (RMDQ) in 11 studies [9,13,26,27,29,30,31,32,33,34], the Oswestry Disability Index (ODI) in one study [35], the pain-related disability (PRD) in one study [36], the Dusseldorf disability scale in one study [37], and the PROMIS—physical function in one study [28]. Intensity of pain was assessed using the Numeric Rating Scale (NRS) in six studies [27,28,29,30,31,37], the Visual Analogue Scale (VAS) in four studies [12,31,35,37], the Brief Pain Inventory (BPI) in two studies [34,35], and the McGill pain Questionnaire Short Form and SF-36 pain scale in two studies [25,26]. Quality of life was assessed using the Short-Form Health Survey (SF-36) in five studies [25,26,27,29,34], the EQ-5D in two studies [32,34], and the Short Form Health Survey (SF-12) in one study [9]. Fear-avoidance behaviors were assessed using the Fear-Avoidance Belief Questionnaire (FABQ) in two studies [36,38], and the Tampa Scale of Kinesiophobia (TSK) in two studies [29,34]. Psychological disorders were assessed using the Generalized Anxiety Disorder 2-item (GAD-2) in three studies [8,33,37], the Personal Health Questionnaire Depression Scale (PHQ-8) in one study [9], the Hospital Anxiety and Depression Scale (HADS) in one study [34], the Beck’s Depression Inventory (BDI) in two studies [13,31] and the PROMIS—depression in one study [28].

### 3.5. Methodological Evaluation

After the application of the Cochrane risk-of-bias tool, nine studies (56%) were at moderate risk of bias, four studies (25%) were at low risk of bias and three studies were determined to be at high risk of bias (Table 3). 

### 3.6. Effect of Intervention

The meta-analysis analyzed the effectiveness of CBT and MBSR in terms of pain, disability, quality of life, depression and Fear-Avoidance Beliefs compared to controls.

#### 3.6.1. Pain

Pain showed a significant decrease both in CBT and MBSR group compared with the control group, respectively SMD −0.73, 95% CI −1.20 to −0.26, *p* = 0.002 for CBT (Figure 2) and SMD −0.30, 95% CI −0.47 to −0.13, *p* = 0.0005 for MBSR (Figure 3). No significant pain reduction was demonstrated (MD −0.05, 95% CI −0.50 to 0.39, *p* = 0.81) when comparing MBSR and CBT (Figure 4).

#### 3.6.2. Disability

Disability scores demonstrated significant improvements after CBT in comparison with controls (SMD −0.88, 95% CI −1.50 to −0.26, *p* = 0.005) (Figure 5). Instead, the reduction of disability after MBSR was not statistically significant compared to controls (MD −0.71, 95% CI −1.53 to −0.11, *p* = 0.09) (Figure 6).

#### 3.6.3. Quality of Life

Quality of life showed significant improvement in CBT and MBSR group compared to controls, respectively SMD 0.69, 95% CI 0.00 to 1.37, *p* = 0.05 for CBT (Figure 7) and MD 2.84, 95% CI 0.31 to 5.37, *p* = 0.03 for MBSR (Figure 8). Moreover, comparing the two intervention groups, a significant difference in quality of life was shown in favor of MBSR (MD 2.54, 95% CI 0.84 to 4.24, *p* = 0.003) (Figure 9).

#### 3.6.4. Depression

Depression scales did not show significant differences between the groups. More precisely, depression did not report statistical improvements between CBT and controls (SMD −0.26, 95% CI −0.72 to 0.19, *p* = 0.26) (Figure 10), MBRS and controls (SMD −1.55, 95% CI −4.53 to 1.43, *p* = 0.31) (Figure 11), and also MBRS and CBT (SMD 0.00, 95% CI −0.27 to 0.27, *p* = 1.00) (Figure 12).

#### 3.6.5. Fear-Avoidance Beliefs

The meta-analysis demonstrated lower fear-avoidance beliefs in patients who underwent CBT compared to control group (SMD −2.17, 95% CI −4.22 to −012, *p* = 0.04) (Figure 13).

#### 3.6.6. Days without Pain

Finally, the number of days without LBP increased in CBT group compared to controls, but without statistical significance (OR 1.38, 95% CI 0.73 to 2.61, *p* = 0.32) (Figure 14).

## 4. Discussion

The link between psychological factors and CLBP has been widely demonstrated in several studies. The aims of this Systematic Review and Meta-Analysis were (1) to identify and describe the most frequently used psychological approaches to treat patients affected by CLBP, and (2) to study the effectiveness of these approaches in terms of reduction of pain, disability, fear-avoidance behaviors, anxiety, depression, and of increase in quality of life. According to the literature [16,19,21], the most common psychological approaches used to treat CLBP are cognitive behavioral therapy and mindfulness-based intervention. CBT demonstrated its effectiveness for different chronic pain conditions [7]. This approach helps patients with maladaptive emotions, behaviors, and cognitions through a goal-oriented and systematic process. CBT was initially used to treat disorders like insomnia, anxiety, and depression, and was later implemented to manage chronic pain [39]. The CBT intervention consists in several sessions guided by a skilled therapist, with different frequency and duration. In these sessions, activities like pain education, relaxation training, managing of automatic thoughts, stress reduction, problem solving and sleep education [39] are performed. MBSR is also becoming increasingly popular and available in the United States [9]. With this treatment, patients are educated about the psychophysiology of stress and are provided opportunities to apply MBSR skills to specific situations [40]. This approach has several contemporary interpretations, based on formal and informal systematic meditation training, patient education, yoga exercises, and individual and group dialogue [41].

The findings of this systematic review and meta-analysis suggest that CBT and MBSR interventions were both associated with an improvement in terms of pain intensity and quality of life when singularly compared to usual care. Disability also improved in both groups when compared to usual care, although it was only statistically significant in patients treated with CBT, which may be due to the paucity of studies that analyzed MBSR intervention. Significant differences in fear-avoidance beliefs were noted in the CBT group compared to usual care. However, no studies analyzed this outcome for the MBSR approach. No meaningful results were noted for depression in both MBSR and CBT groups. Moreover, only two RCT compared CBT to MBSR, showing no significant improvements in pain intensity and depression along with a better quality of life for the MBSR intervention [9,28].

Another meta-analysis [20] studied several psychosocial interventions to treat patients affected by CLBP. This study demonstrated an improvement in pain, QoL and work-related disability in the intervention group towards the waiting list group.

Our results agree with those by Gotink et al. [22], who studied MBSR applied to chronic illness. Their review shows the large use of this treatment with patients affected by cancer, cardiovascular diseases, mental disorders, and non-specific chronic pain. Indeed, an improvement in depressive symptoms and physical health and a decrease in pain burden, intensity and disability are reported in patients affected by non-specific chronic pain.

Regarding the implementation of CBT, Richmond et al. [19] shows its effectiveness for non-specific low back pain, with improvement of pain, functional disability, and quality of life. In addition, Morley et al. [42] show the improvement of pain and functional disability in patients with chronic pain.

Psychological factors in people affected by LBP are associated with increased risk of developing disability [43]. For instance, the symptom of depression and the catastrophizing of pain predict poor low back pain-related outcomes [44,45]. Therefore, cognitive and emotional factors have a crucial impact on pain perception and, in line with the literature [46], it is fundamental to identify and take care of psychological factors, through a multidisciplinary approach in patients with chronic pain. Indeed this approach should be considered because each aspect requires specific interventions [47].

This review has several limitations. Firstly, there is some difference in heterogeneity between studies regarding CBT and MBSR. In particular, the studies involving CBT presented high heterogeneity due to the greater number of studies included, the different types of CBT performed, the different duration of interventions, and the tools used. Instead, the studies investigating MBSR used the same tools for the analyzed outcomes, resulting therefore in lower heterogeneity. Additionally, demographic characteristics of the participants were different in the included studies, with various gender and age distribution. However, this did not influence the statistical analysis of the studies. There are differences regarding the quality of the studies; indeed, the major number of the studies included were of moderate quality (*n* = 9), whereas four studies had a low risk of bias, and three studies had a high risk of bias. We also decided to include in the meta-analysis the studies with a high risk of bias, which is another limitation regarding the number of studies included. The participants of most studies, except for three studies [25,26,27], were not blind to treatment allocation, given the nature of the intervention. Another limitation of this study is the heterogeneity of the types of treatment in the control groups. Indeed, we found different types of treatments in the control groups such as physiotherapy, physical exercises, educational programs and drug treatments. In only two studies we found the waiting list, and we suppose it may be the most adequate control group for the reduction of bias, hence the need to develop RCTs with waiting lists as a control group for an appropriate analysis of the effectiveness of psychological interventions. To reduce the heterogeneity in the analysis of outcomes as pain, it would be appropriate to develop RCTs that use the same tools.

## 5. Conclusions

In the present study we analyzed the most used and effective psychological approaches to treat patients affected by chronic LBP. CBT and MBSR have proven their significant effectiveness to improve pain intensity and quality of life compared to controls. These approaches also demonstrated their efficacy in reducing disability and fear-avoidance, but without significant results. The importance of treating psychological aspects is widely proven, but the paucity and heterogeneity of the studies included cannot make us confident to affirm which is the most effective treatment. Further studies are needed to compare CBT and MBSR.

## Figures and Tables

**Figure 1 ijerph-19-00060-f001:**
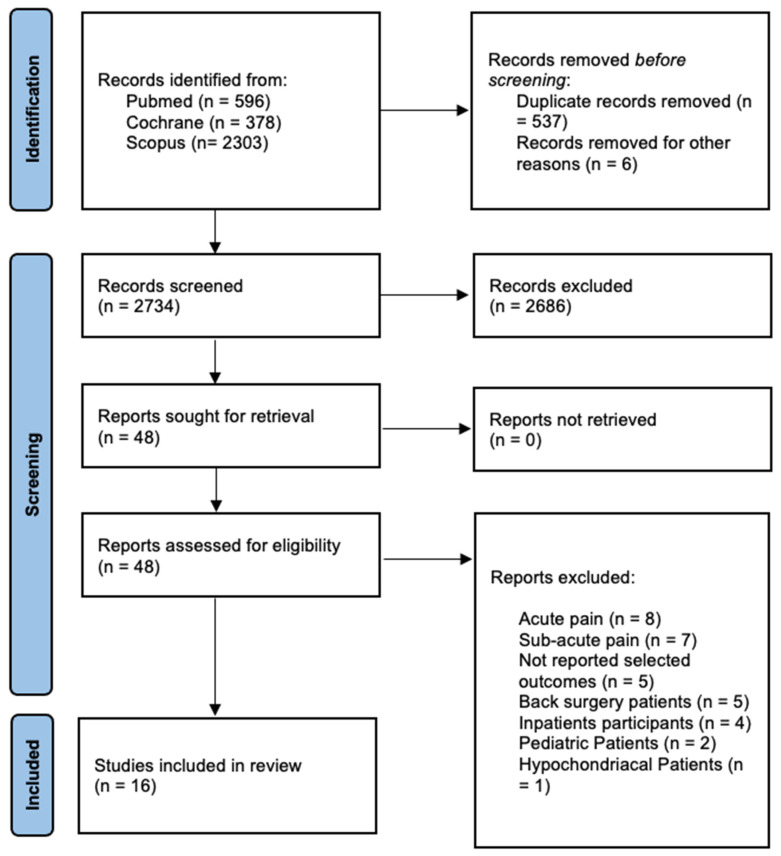
Preferred reporting items for systematic review and meta-analysis (PRISMA 2020).

**Figure 2 ijerph-19-00060-f002:**
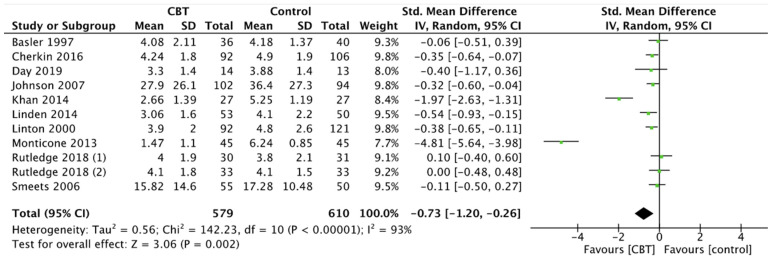
Pain: CBT versus control.

**Figure 3 ijerph-19-00060-f003:**
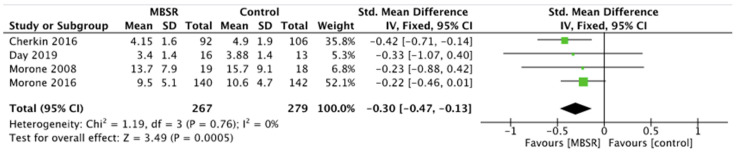
Pain: MBSR versus control.

**Figure 4 ijerph-19-00060-f004:**
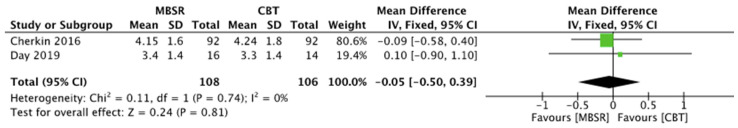
Pain: MBSR versus CBT.

**Figure 5 ijerph-19-00060-f005:**
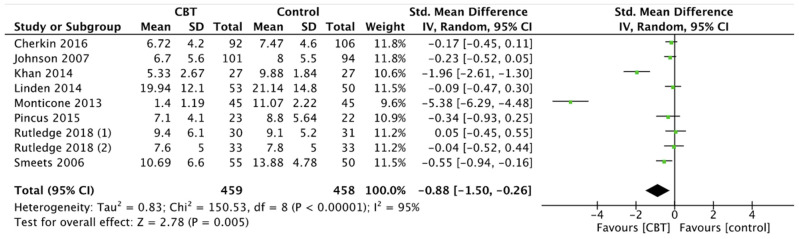
Disability: CBT versus control.

**Figure 6 ijerph-19-00060-f006:**
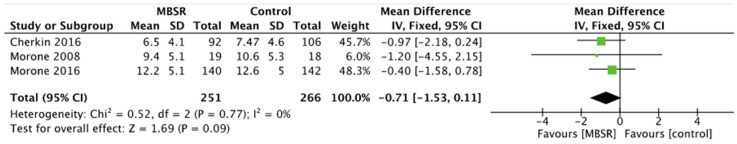
Disability: MBSR versus control.

**Figure 7 ijerph-19-00060-f007:**
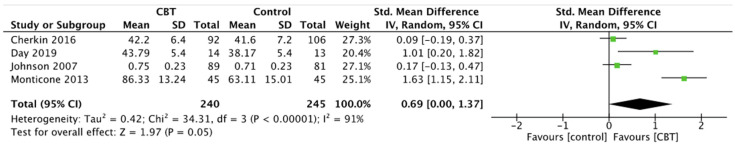
Quality of Life: CBT versus control.

**Figure 8 ijerph-19-00060-f008:**
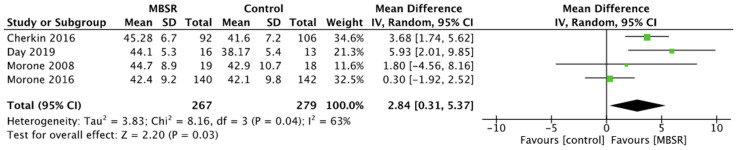
Quality of Life: MBSR versus control.

**Figure 9 ijerph-19-00060-f009:**
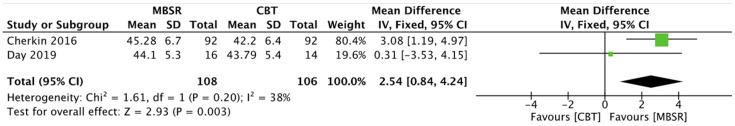
Quality of Life: MBSR versus CBT.

**Figure 10 ijerph-19-00060-f010:**
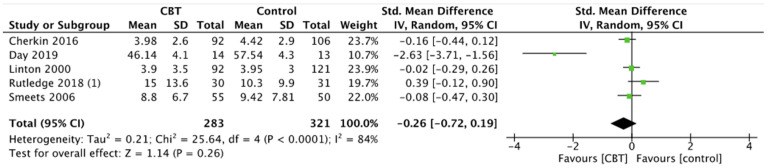
Depression: CBT versus control.

**Figure 11 ijerph-19-00060-f011:**
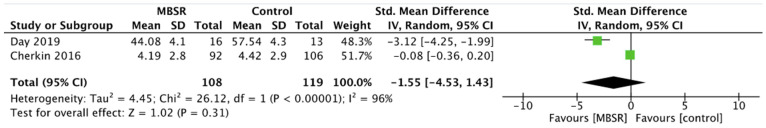
Depression: MBSR versus control.

**Figure 12 ijerph-19-00060-f012:**
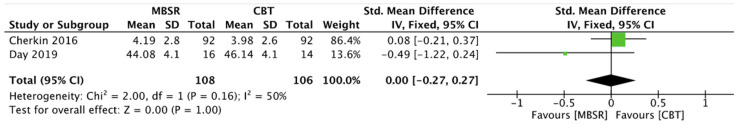
Depression: MBSR versus CBT.

**Figure 13 ijerph-19-00060-f013:**
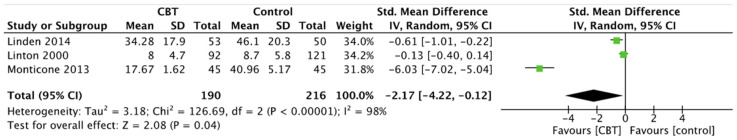
Fear-Avoidance beliefs: CBT versus control.

**Figure 14 ijerph-19-00060-f014:**
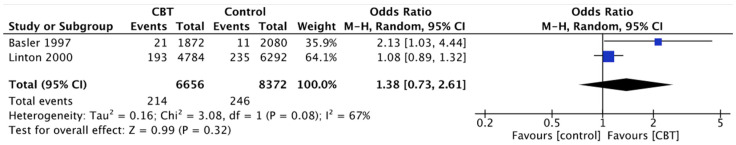
Days without pain: CBT versus control.

**Table 1 ijerph-19-00060-t001:** Main characteristics of the included studies and samples.

Author	Year	Country	Study Group			Control Group		
			N.	Age	Sex	N.	Age (Years)	Sex
Cherkin et al.	2016	USA	116	50 ± 11.9	71% F 29% M	113	48.9 ± 12.5	87% F 13% M
			112	49.1 ± 12.6	66% F 34% M			
Monticone et al.	2013	Italy	45	49 ± 8	60% F 40% M	45	49.7 ± 7	55% F 45% M
Johnson et al.	2007	UK	116	47.3 ± 10.9	61% F 39% M	118	48.5 ± 11.4	58% F 42% M
Smeets et al.	2006	The Netherlands	58	42.5 ± 9.7	58.6% F 41.4% M	53	42.7 ± 9.1	41.5% F 58.5% M
			61	40.7 ± 10.1	37.7% F 62.3% M	51	40.5 ± 11.2	37.3% F 62.7% M
Rutledge et al.	2018	USA	30	62.5 ± 11.3	13% F 87% M	31	64.3 ± 12.7	6% F 92% M
Rutledge et al.	2018	USA	33	54 ± 14.8	37.5% F 62.5% M	33	52.6 ± 12.5	39.4% F 60.6% M
Linden et al.	2014	Germany	53	50.4 ± 6.9	68% F 32% M	50	49.7 ± 7	68% F 32% M
Khan et al.	2016	Pakistan	27	39.61 ± 5.3	54% F 46% M	27	39.61 ± 5.3	54% F 46% M
Pincus et al.	2015	UK	45	43.7 ± 16.3	60% F 40% M	44	45.4 ± 15.8	38.6% F 61.4% M
Basler et al.	1997	Germany	36	49.3 ± 9.7	75.6% F 24.4% M	40	49.3 ± 9.7	75.6% F 24.4%M
Linton et al.	2000	Sweden	107	44	70% F 30% M	70	45	71% F 29% M
						66	44	74% F 26% M
Zgierska et al.	2016	USA	21	51.8 ± 9.7	80% F 20% M	14	51.8 ± 9.7	80% F 20% M
Morone et al.	.2008	USA	19	74.1 ± 6.1	53% F 47% M	18	75.6 ± 5	61% F 39% M
Morone et al.	2009	USA	16	78 ± 7.1	69% F 31% M	19	73 ± 6.2	58% F 42% M
Morone et al.	2016	USA	140	75 ± 7.2	66% F 34% M	142	74 ± 6.0	66% F 34% M
Day et al.	2019	USA	23	49.9 ± 11.9	61% F 39% M	23	48.1 ± 16.1	52% F 48% M
						23	54.3 ± 14.9	44% F 56% M

**Table 2 ijerph-19-00060-t002:** Clinical results of the included studies.

Study	Intervention (s)	Control	Follow-Up	Outcomes (Tool)	Conclusion
Cherkin et al., 2016	Mindfulness: body scan, yoga, meditation, for 8 weeks. CBT: education about chronic pain, relationships between thoughts and emotional and physical reactions, sleep hygiene, setting and working toward behavioral goals, relaxation skills, activity pacing, and pain-coping strategies, for 8 weeks	Usual care (whatever care participants received)	12 months	Disability (RMDQ) QoL (SF-12) Depression (PHQ-8) Anxiety (GAD-2)	Among adults with CLBP, treatment with MBSR or CBT, compared with usual care, resulted in greater improvement in back pain and functional limitations at 26 weeks, with no significant differences in outcomes between MBSR and CBT
Monticone et al., 2013	CBT: intervention to modify fear of movement beliefs, catastrophizing thinking, and negative feelings, and ensuring gradual reactions to illness behaviors, for 5 weeks	Active and passive mobilizations of the spine, and exercises aimed at stretching and strengthening muscles, and improving postural control, for 5 weeks	12 months	Disability (RMDQ) Pain (NRS) QOL (SF-36) Fear advoidance behaviours (TSK)	The long-lasting multidisciplinary program was superior to the exercise program in reducing disability, fear- avoidance beliefs and pain, and enhancing the quality of life of patients with chronic low back pain. The effects were clinically tangible and lasted for at least 1 year after the intervention ended.
Johnson et al., 2007	CBT: educational pack containing a booklet and audio-cassette + problem solving, pacing and regulation of activity, challenging distorted cognitions about activity and harm, for 6 week	Educational pack containing a booklet and audio-cassette + usual care for 6 weeks	15 months	Pain (VAS) Disability (RMDQ) QoL (EQ-5D)	CBT intervention program produces only modest effects in reducing LBP and disability over a 1-year period.
Smeets et al., 2006	CBT: operant behavioral graded activity training and problem solving training Active Physical Treatment (APT): aerobic training, and three dynamic static strengthening exercises for 4 weeks Combined Treatment (CT): CBT + APT	Waiting List (WL) for 10 weeks	12 months	Disability (RMDQ) Pain (VAS) Depression (BDI)	CBT are as effective in reducing the subjective experienced level of functioning
Rutledge et al., 2018	CBT: to provide core educational information, guide patients’ learning and skills development, and structure self-monitoring exercises for the respective session, for 8 weeks	Supportive Care: - Education by distribution of a standard text - Active Listening by the therapist to participant’s concerns - Supportive care following Rogerian principles	12 months	Disability (RMDQ) Pain (NRS) Depression (BDI)	No evidence of meaningful effect size differences between the treatments.
Rutledgeet al., 2018	CBT: managing pain, managing stress, thinking differently, assertive communication, setting goals for 8 weeks	Supportive Care: - Education by distribution of a standard text - Active Listening by the therapist to participant’s concerns - Supportive care following Rogerian principles	12 months	Disability (RMDQ) Pain (NRS)	CBT versus SC therapy demonstrated statistically significant and comparable patterns of improved outcomes on measures of back pain disability, pain severity, and self- rated improvement.
Linden et al., 2014	general orthopedic inpatient treatment + therapy in reference to the GRIP and the pain and illness management program from Geissner at al. with additional cognitive behavior therapy interventions which aim at stress reduction and problem solving, self monitoring, pain management, change in dysfunctional cognitions, reduction of avoidance behavior, and wellbeing therapy for 3 weeks	General orthopedic inpatient treatment	3 weeks	Fear advoidance behaviours (FABQ) Pain (VAS) Pain related disability (PDI)	CBT can reduce back pain and increase functional coping, and that this is not mediated by an improvement in mental health and a reduction of depression, anxiety or somatization in general or by induc- tion of some general optimistic views. Pain is not identical with mental problems.
Khan et al., 2016	general exercise + CBT aimed to guide patients to achieve their daily life goals. CBT consisted of operant behavioural graded activity and problem solving training, for 12 weeks	General exercise at home 2 times per day and at least 5 times a week (for 12 weeks)	12 weeks	Disability (RMDQ) Pain (VAS)	This study found that both CBT with General exercises and General exercises alone significantly reduced pain intensity and disability in patients with chronic low back pain. Furthermore, subjects treated with CBT & Exercises showed an additional clinical benefit as compared to General Exercises only. Hence, CBT & Exercises could be a better option in clinical practice.
Pincus et al., 2015	Session content was not structured, and at the discretion of therapists, included any features of Contextual Cognitive-Behavioural Therapy (CCBT) they thought were appropriate at the point with that patient.	Physiotherapy, comprised back to fitness group exercises with at least 60% of content exercise-based.	3 months	Fear advoidance behaviours (TSK) pain (Brief Pain Inventory) disability (RMDQ) anxiety and depression (HADS) QoL (EQ-5D and SF-36)	CCBT is a credible and acceptable intervention for LBP patients who exhibit psychological obstacles to recovery.
Basler et al., 1997	medical treatment such as pain medication, nerve blocks, TENS, and physical therapy + CBT therapy: education, relaxation, Modifying thoughts and feelings, Pleasant activity scheduling, Training of posture	Medical treatment such as pain medication, nerve blocks, TENS, and physical therapy	6 months	Pain (NRS) Disability (Dusseldorf disability scale)	Experimental subjects reported less pain, more pleasurable activities and feelings, less avoidance and less catastrophizing, and disability was reduced. The results were maintained at follow-up. Patients who only received medical treatment showed little improvement. Data indicate that the program meets the needs of the patients and should be continued.
Linton et al., 2007	Sessions were organized to activate participants and promote coping. Each session began with a short review, in which homework was covered. The treatment lasts 6 weeks	1. pamphlet: straightforward advice about the best way to cope with back pain by remaining active and thinking positively. 2. Information package: advice and illustrations showing how the patient might cope with spinal pain or prevent it by such methods as lifting properly and main- taining good posture.	12 months	Pain (VAS) Depression and anxiety (HAD) Fear Advoidance (FABQ)	This study demonstrates that CBT group intervention can lower the risk of a long-term disability developing.
Zgierska et al., 2016	Usual care and opioid therapy management + manualized training in the meditation-CBT intervention 2 h per week for 8 weeks	Pharmacotherapy, opioid therapy management and physical therapy	26 weeks	Pain (Brief Pain Inventory) Disability (ODI)	Mindfulness meditation and CBT-based interventions have the potential to safely reduce pain severity and sensitivity in patients with opioid-treated CLBP
Morone et al., 2008	Mindfulness: body scan, sitting practice, walking meditation	Waiting List	3 months	Pain (McGill pain Questionnaire- Short Form and SF-36 pain scale) Disability (RMDQ) QoL (SF-36)	The mindfulness intervention sustained improvement in physical function and pain acceptance.
Morone et al., 2009	Mindfulness: body scan, sitting practice, walking meditation	Educational program (8 weeks), including lectures, group discussion, and homework assignments based on the health topics discussed	4 months	Disability (RMDQ) Pain (McGill pain Questionnaire- Short Form and SF-36 pain scale) QoL (SF-36)	A mindfulness meditation program and an education control group both showed improvement at program completion on measures of pain, and physical and psychological function.
Morone et al., 2016	Mindfulness: body scan, sitting practice, walking meditation for 8 weeks	Educational program on a successful aging curriculum known as the 10 Keys to Healthy Aging	6 months	Disability (RMDQ) Pain (NRS) QoL (SF-36)	A mind-body program for chronic LBP improved short-term function and long-term current and most severe pain. The functional improvement was not sustained.
Day et al., 2019	MBCT for pain protocol integrates cognitive and be- havioral techniques with mindfulness-based strategies	CT techniques delivered: treatment involved traditional Beckian style column technique restructuring exercises Mindufulness: cognitive-behavioral and mindful movement components removed	6 months	Pain (NRS) Physical function (PROMIS) Depression (PROMIS)	The findings show that MBCT is a feasible, tolerable, acceptable, and potentially efficacious treatment option for CLBP. Further, MBCT, and possibly CT, could have sus- tained benefits that exceed MM on some important CLBP outcomes.

**Table 3 ijerph-19-00060-t003:** Cochrane risk-of-bias tool for randomized controlled trials.

	Random Sequence Generation	Allocation Concealment	Blinding (Participants and Personnel)	Blinding (Outcome Assessment)	Incomplete Outcome Data	Selective Reporting	Other Bias	Risk of Bias
Cherkin et al., 2016	L	L	H	L	L	L	H	U
Monticone et al., 2013	L	L	H	L	L	L	L	L
Johnson et al., 2007	L	L	H	H	L	L	H	U
Smeets et al., 2006	L	L	H	L	L	L	H	U
Rutledge et al., 2018	L	L	H	L	L	L	H	U
Rutledge T et al., 2018	L	L	H	H	L	L	L	U
Linden et al., 2014	L	U	H	L	L	L	H	U
Khan et al., 2016	L	U	H	L	L	L	H	U
Pincus et al., 2015	L	L	H	U	L	L	H	U
Basler et al., 1997	L	L	H	L	L	L	H	U
Linton et al., 2000	L	U	H	U	L	L	H	H
Zgierska et al., 2016	L	U	H	H	L	L	H	H
Morone et al., 2008	L	U	U	H	L	L	H	H
Morone et al., 2009	L	L	U	L	H	L	L	L
Morone et al., 2016	L	L	U	L	L	U	L	L
Day et al., 2019	L	L	H	L	L	L	L	L

L: low; U: unclear; H: high.

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
