# Peer review of "Psychological Approaches for the Integrative Care of Chronic Low Back Pain: A Systematic Review and Metanalysis"

_ijerph, 2021, doi:10.3390/ijerph19010060_

Round 1
Reviewer 1 Report
This systematic review included randomized trials that evaluated the effect of psychological approaches on patients suffering from chronic low back pain. A total of 16 studies were included and two main approaches were identified across the studies: cognitive behavioral therapy and mindfulness-based intervention. It demonstrated that both approaches were associated with an improvement in terms of pain intensity and quality of life when compared to usual care.
Two major concerns:
- Previous studies have investigated the relationship between psychological approaches on lower back pain. Current study will need to justify its novelty and necessity of another systematic review, including differences/similarities compared to previous ones. Brief search reviewed the previous systematic reviews: 1. Meta-analysis of psychological interventions for chronic low back pain. 2007. Health Pyschol 26(1):1-9. 2. The effectiveness of cognitive behavioural treatment for non-specific low back pain: a systematic review and meta-analysis. 2015 Plos One 10(8):e0134192.
- As illustrated by Table 3, the risk of bias was significant across the studies, especially three studies were determined to be a high risk of bias by authors. It might be appropriate to exclude those studies of low quality/high bias from the meta-analysis.
Author Response
Dear Reviewer,
Thank you for the attention and the time devoted to our manuscript. The responses to their comments and suggestions are presented below in blue colour. The changes within the revised manuscript are made evident by red colour.
Kind regards,
The authors

Reviewer 2 Report
As a general remark I would like to stress the fact that the type of interventions is not well defined. What is usual care? Exercise in control groups : intensity?
The intervention period is from some weeks to some months…
The conclusion should mention with more stress this heterogenicity, any way of paying attention to pain patients can influence pain patients. Give a suggestion for future research, how to contorl these parameters?
1. The manuscript is an attempt to review the effects of behavioural / psychological interventions in chronic LBP patients frpm the last 25 years.
2. Strengths : the amount of published papers is huge, the attempt to review all this and trying to organize in a way to extract statistical information is an enormous work I think.
Line 133 : 3277 articles reviewed by reading the abstracts by 2 authors to reduce to 48 articles. How long did this take? Weaknesses: the definition of "usual care" (line 68/69) or exercise or another intervention is unclear, I am a clinical practitioner in physical medicine with 80% out-patient care, a lot of LBP patients, since >30 years we treat patients in a "multi disciplinary" approach, this means that there is a physical part and a psychological approach. This is different in "diffuse chronic pain" because of the more localized symptoms. I think that there is a big bias in the treatment descriptions of control groups as well as in the intervention groups, the spread in demographics shows this already (line 143/144 ) In line 150 : 13/80 % females in experimental group and 6 to 87% in control group. I doubt that outcomes can be compared.
3 and 4 :
Table 2 line 161 The intervention CBT is sometimes defined as "exercise" and in another study "exercise" is classified in the control group. Interventions as short as 5 weeks in the Montilone 2013 study, 3 weeks in Linden 2014 ... The only control group, in my opinion, can be patients on a waiting list with no intervention at all.
Line 191: the evaluation of "pain", in table I see the Cherkin, 2016 study where the conclusion is: " a greater improvement in back pain" but when I check the "outcomes" of this study in table 2 there is NO pain scale mentioned.
Line 275.... The aim of the study : (1) to identify 275 and describe the most frequently used psychological approaches to treat patients affected 276 by CLBP, and (2) to determine which psychological approach is most effective in terms of 277 pain, disability, fear avoidance behaviors, anxiety, depression, and quality of life.
I do not think that the analysis presented gives a response to these aims. That is why I think the conclusions should be formulated in a different way
Author Response

(The authors gave the same response as above.)

Reviewer 3 Report
As one who has suffered from chronic low back pain for my entire adult life, I read this review as both a scientist and an interested consumer. I found it to be well-written, mostly clear, and rigorous in methodology. I see only one major point where the manuscript could be improved/clarified.
It is not clear whether the study involves a confound between CBT/MBSR and possible increases in the amount of beneficial physical exercise that those in the therapy groups might have been encouraged to undertake as a result of their improved outlooks. One would expect there to be an additive effect between CBT/MBSR (although a lesser effect it seems) and physiological manipulations. Indeed, lines 314-317 appear to confirm this possibility. Did the CBT patients participate in any physical therapy activities at all? Related to this, the control groups were sometimes listed as “usual care.” What does this involve?
The possibility of an interaction between physical and psychological manipulations is, of course crucial to the interpretation of the outcomes. If there was not an interaction, one has to suppose that descending cortical inhibition of lumbar pain pathways may have played a role in pain relief. If not, one would expect the improved psychological outlook-especially in the area of fear avoidance, might have encouraged the CBT/MBSR patients to participate in beneficial physical activities. For chronic LBT sufferers, either is fine if it provides relief!
Minor comments.
Table 20, Smeets et al., ”Personal experienced level of functioning”..not clear- do the authors mean “subjective” experience?
Line 68. “phycological” typo?
Line 85. “Comparation” does not appear in my dictionary search. “Comparison”?
Author Response
Dear Reviewer,
Thank you for the attention and the time devoted to our manuscript. The responses to your comments and suggestions are presented below in blue colour. The changes within the revised manuscript are made evident by red colour.
Kind regards,
The authors
